# Retrospective CT/MRI Texture Analysis of Rapidly Progressive Hepatocellular Carcinoma

**DOI:** 10.3390/jpm10030136

**Published:** 2020-09-21

**Authors:** Charissa Kim, Natasha Cigarroa, Venkateswar Surabhi, Balaji Ganeshan, Anil K. Pillai

**Affiliations:** 1Department of Surgery, Huntington Memorial Hospital, 100 W California Blvd, Pasadena, CA 91105, USA; charissa.kim7@gmail.com; 2Department of Diagnostic and Interventional Imaging, McGovern Medical School at UTHealth, 6431 Fannin St, Houston, TX 77030, USA; natasha.cigarroa@uth.tmc.edu; 3Institute of Nuclear Medicine, University College Medicine, 5th Floor, Tower University College Hospital, 235 Euston Road, London NW1 2BU, UK; b.ganeshan@ucl.ac.uk; 4Division of Vascular Interventional Radiology, The University of Texas Southwestern Medical Center, 5323 Harry Hines Blvd, Dallas, TX 75390, USA; Anil.Pillai@UTSouthwestern.edu

**Keywords:** hepatocellular carcinoma, feature detection, qualitative visual analysis, texture analysis

## Abstract

Rapidly progressive hepatocellular carcinoma (RPHCC) is a subset of hepatocellular carcinoma that demonstrates accelerated growth, and the radiographic features of RPHCC versus non-RPHCC have not been determined. The purpose of this retrospective study was to use baseline radiologic features and texture analysis for the accurate detection of RPHCC and subsequent improvement of clinical outcomes. We conducted a qualitative visual analysis and texture analysis, which selectively extracted and enhanced imaging features of different sizes and intensity variation including mean gray-level intensity (mean), standard deviation (SD), entropy, mean of the positive pixels (MPP), skewness, and kurtosis at each spatial scaling factor (SSF) value of RPHCC and non-RPHCC tumors in a computed tomography (CT) cohort of n = 11 RPHCC and n = 11 non-RPHCC and a magnetic resonance imaging (MRI) cohort of n = 13 RPHCC and n = 10 non-RPHCC. There was a statistically significant difference across visual CT irregular margins *p* = 0.030 and CT texture features in SSF between RPHCC and non-RPHCC for SSF-6, coarse-texture scale, mean *p* = 0.023, SD *p* = 0.053, MPP *p* = 0.023. A composite score of mean SSF-6 binarized + SD SSF-6 binarized + MPP SSF-6 binarized + irregular margins was significantly different between RPHCC and non-RPHCC (*p* = 0.001). A composite score ≥3 identified RPHCC with a sensitivity of 81.8% and specificity of 81.8% (AUC = 0.884, *p* = 0.002). CT coarse-texture-scale features in combination with visually detected irregular margins were able to statistically differentiate between RPHCC and non-RPHCC. By developing an image-based, non-invasive diagnostic criterion, we created a composite score that can identify RPHCC patients at their early stages when they are still eligible for transplantation, improving the clinical course of patient care.

## 1. Introduction

Hepatocellular carcinoma (HCC) is the second leading cause of cancer-related mortality worldwide, with a 5-year survival rate of <15% and propensity for progression and metastasis [1]. While hepatectomy and transplantation can be curative in early-stage HCC, only 20–30% of patients with HCC are candidates for these early interventions [2]. Indeed, liver transplantation has been shown to result in high rates of recurrence-free survival specifically for early-stage, unresectable HCC patients [3]. As the allocation of transplant livers is limited by a longstanding organ shortage, patients with HCC are triaged for liver transplantation using a modified version of the Model for End-Stage Liver Disease (MELD) score that affords MELD exception points if patients meet the Milan Criteria of one lesion greater than or equal to 2 cm and less than or equal to 5 cm in size or two or three lesions each greater than or equal to 1 cm and less than or equal to 3 cm in size [4]. As a result, it is clinically crucial to identify and characterize HCC tumors at their initial stages when the patients are still eligible for transplantation.

Patients with T1 HCC (one lesion less than 2 cm) are currently not eligible for priority listing for liver transplantation. A common practice is to wait without locoregional therapy until tumor growth occurs from T1 to T2 (one lesion between 2–5 cm or 2–3 lesions less than 3 cm) to be eligible for listing with Model for End-Stage Liver Disease Exception [5]. This “wait and not treat” approach for T1 HCCs until tumor growth to T2 HCC is reasonable for most patients; however, the growth rate and tumor volume doubling time (TVDT) of early-stage HCCs are variable with a reported TVDT range of 11–851.2 days [6].

Although most HCC tumors display indolent growth, a particularly aggressive subtype of HCC has been identified that demonstrates rapid progression: rapidly progressive HCC (RPHCC) [7,8]. During the “wait and not treat” period, the RPHCC tumors grow rapidly and may become ineligible for transplant. The distinctive radiographic features of RPHCC versus non-RPHCC have been difficult to determine. This is partially because previous studies have not investigated RPHCC imaging features at the radiomic depth necessary to quantitatively differentiate RPHCC from non-RPHCC. Of note, texture analysis is a quantitative method that assesses the relative distribution of pixel or voxel gray levels in images, providing objective measures of features within CT and MRI images [9]. By capturing pixel or voxel levels of tumor heterogeneity, texture analysis has the potential to extract features from RPHCC tumors that may differ from features in non-RPHCC tumors [10].

Our current understanding of RPHCC is limited by cohort size and by an absence of comprehensive data regarding the specific imaging features that distinguish RPHCC from non-RPHCC tumors. To address these gaps in knowledge and in light of the clinical benefits of early RPHCC detection and treatment, we conducted a retrospective study with the objective of determining if baseline radiologic features and texture analysis can aid in the accurate detection of RPHCC.

## 2. Materials and Methods

An institutional review board waiver was obtained for this retrospective analysis. In this unblinded retrospective study, baseline CT and MR images were reviewed from the REDCap Database (database of cases discussed by the Liver Tumor board at our institution) by an experienced radiologist to establish rapid progression, which is defined as a lesion that exhibited the following growth criteria: 50% growth in <3 months, 100% growth in <6 months, 150% growth in <12 months, or 200% growth in <15 months. Of 700 cases from the REDCap tumor databases, RPHCC (n = 23) and control (n = 21) cases were selected based on inclusion/exclusion criteria for the purpose of this study. Inclusion criteria comprised of the following: (1) lesions classified as Hepatocellular Carcinoma LI-RAD 5 on LI-RAD v2018; (2) images available for baseline and follow up, (3) images with both arterial and venous phases (Figure 1).

LI-RAD 5 v2018 criteria was designated by non-rim arterial hyperenhancement and nonperipheral wash-out or threshold growth if 10–19 mm, or ≥2 of the additional major features (enhancing capsule, nonperipheral washout, and threshold growth) if 10–19 mm, or ≥1 of the additional major features if ≥20 mm. Tumor size was recorded as the maximum tumor dimension in the axial plane on arterial phase images, and each patient only had one tumor. Exclusion criteria comprised of the following: (1) lesions that were ablated during the time interval; (2) patients who received chemotherapy; (3) baseline lesions less than 1 cm; and (4) poor quality MRI/CT images (Figure 1). Each patient only had one tumor.

### 2.1. MR Technique

MRI was performed with extracellular contrast agents (gadobenate dimeglumine, 0.1 mmol/kg) with a dynamic contrast-enhanced liver protocol performed on 1.5-T (Ingenia, Philips Healthcare; Signa HD, GE Healthcare, Science Park, Amsterdam, The Netherlands) or 3-T (Ingenia, Philips Healthcare; Discovery MR750, GE Healthcare) imaging platforms. Early arterial phase images were obtained 20–25 s after gadolinium-based IV contrast administration; late arterial phase images were obtained 35–45 s after contrast administration; portal venous phase images were obtained 80–90 s after contrast administration; and delayed phase images were obtained 3 min after contrast administration.

### 2.2. CT Technique

Contrast-enhanced CT examinations were performed with a multidetector CT (LightSpeed 16, GE Healthcare, Milwaukee, WI, USA; Sensation 64 and 128, Siemens Healthineers). The CT parameters were 120 kVp, 200–300 mAs, 1.25 mm × 16 and 0.625 mm × 64/128 section collimation (pitch, 0.6–0.938), and a single-breath-hold helical acquisition of 5–8 s depending on the liver size. An initial nonenhanced scan was acquired with 3-mm section thickness through the liver. After a 20-gauge intravenous cannula was inserted, a total amount of non-ionic iodinated contrast agent (Omnipaque; Nycomed, Princeton, NJ, USA) administered was determined based on body weight (2 mL/kg), and it was injected with an automatic injector at a rate of 4–5 mL/sec. Hepatic arterial phase imaging was initiated with the bolus tracking technique (120 kVp; 40–60 mA; monitoring frequency from 12 s after the contrast injection, 1 s; trigger threshold, 100 HU in the descending aorta; delay from trigger to the initiation of CT data acquisition, 15 s). Portal and delayed phase images were obtained 60–70 and 180 s after the initiation of the injection of the contrast material, respectively.

### 2.3. Study Population

The included cases were separated based on imaging modality CT versus MRI and non-RPHCC (Figure 2A) versus RPHCC classification (Figure 2B,C). This resulted in a CT cohort of n = 11 RPHCC and n = 11 non-RPHCC and an MRI cohort of n = 13 RPHCC and n = 10 non-RPHCC. Demographic data, including age, gender, race, comorbidities, and Child-Pugh Class are presented in Table 1.

### 2.4. Qualitative Visual Analysis

A single radiologist with >10 years of experience assessed for features of baseline images, and the following characteristics were recorded: tumor margins, necrotic component, internal vascularity, washout, pseudo-capsule, and arterial phase enhancement. The margin was considered to be well-defined if >90% of the entire tumor circumference was ‘pencil-thin’ sharp when viewed on arterial, portal venous, or delayed phase images using a narrow window setting.

### 2.5. Texture Analysis

Texture analysis (TA) was performed by a single-trained reader under the supervision of one experienced abdominal radiologist (10 years). Representative single-slice images at the level of the largest diameter of LIRADS v2018 5 lesion in the arterial phase were identified and were then sent to a commercially available texture analysis research software platform. Texture features were extracted using commercially available research software (TexRAD version 3.9, provided by Feedback Medical Ltd., Cambridge UK—https://fbkmed.com/texrad-landing/). Using the software, a region of interest (ROI) was manually drawn on a single slice corresponding to the largest diameter of the lesion in the arterial phase for RPHCC (Figure 3A–D) and non-RPHCC (Figure 4A–D) patients. CT/MR TA assessed the heterogeneity within each ROI on the CT and MR axial images using a filtration-histogram based texture analysis technique. A filtration step using a band-pass Laplacian of Gaussian (LoG) filter (similar to non-orthogonal wavelet) comprised of extracting and enhancing image features of different sizes and intensity variation corresponding to the spatial scale of the filter (SSF in radius). The feature scales used ranged from SSF = 2 to 6 mm where a fine-texture scale corresponded to SSF = 2 mm, medium-texture scales corresponded to SSF = 3–5 mm, and a coarse-texture scale corresponded to SSF = 6 mm. Following the filtration step, the quantification of texture was undertaken using statistical- and histogram-based parameters at each derived (filter scale, SSF value) image as well as on the conventional image (without filtration, SSF = 0). Statistical and histogram parameters comprised of mean intensity (which reflects average brightness), standard deviation (which reflects the width of the histogram or dispersion from the average), entropy (which reflects irregularity), mean of positive pixels (which reflects the average brightness of only positive pixel values), kurtosis (which reflects the pointedness or peakedness or sharpness of the histogram distribution), and skewness (which reflects the asymmetry of the histogram distribution) [11]. Filtration histogram-based TA has undergone a qualification and validation process on CT and MRI as evidenced from a number of publications in the literature in oncological and non-oncological settings [9,12,13,14]. One aspect of the qualification and validation process is to understand what the filtration histogram-based TA actually means and how it reflects heterogeneity. A simulation and phantom study by Miles et al. [11] has further highlighted what the filtration-histogram based TA actually means and how it reflects different components of heterogeneity (object/feature size, number/concentration of objects/features, and variation in intensity of the objects/features in relation to the background). These values were recorded for each patient case and subsequently underwent statistical analysis.

### 2.6. Statistical Analysis

To assess the difference in CT TA and MR TA parameters (mean intensity, standard deviation, entropy, mean of positive pixels, skewness, and kurtosis at different SSF values) between rapidly progressive HCC from controls, a non-parametric Mann–Whitney test was employed. Parameters that showed a significant difference were visualized using the box and whisker plots. Receiver operating characteristics (ROC) analysis determined the diagnostic criteria (area under the ROC curve (AUC), optimal cut-off, sensitivity, specificity, and *p*-value) for the significant metrics. A composite score was developed by combining the most significant metrics (texture, other imaging and clinical metrics). A composite score was created by adding each significant metric in its binarized form (0—negative diagnosis and 1—positive diagnosis) based on the respective cut-off established from ROC analysis (Figure 1). A Mann–Whitney test further assessed whether this composite score was significantly different between rapidly progressing HCC and controls and better than the individual metrics. ROC analysis determined the diagnostic criteria (AUC, optimal cut-off, sensitivity, specificity, and *p*-value) for the composite score. Fisher’s exact test was used to compare categorical characteristics within study groups where applicable, and the chi-squared test was used for qualitative visual feature analysis where applicable. A Wilcoxon rank sum test was used for AFP. For any test, a *p* value ≤ 0.05 was considered to be significant. All statistical analyses were performed by using SPSS version 25 (IBM Corp. Released 2017. IBM SPSS Statistics for Macintosh, Version 25.0. Armonk, NY, USA: IBM Corp.).

## 3. Results

### 3.1. Cohort and Visual Features

The study groups of (RPHCC and non-RPHCC) were comparable with no statistically significant differences in age, gender, race, comorbidities, and Child-Pugh Class (Table 1). Of the cases analyzed, 37.5% of all RPHCC lesions displayed irregular margins compared to 0% in the control group (*p* = 0.002). When stratified by imaging modality, CT radiologic features including internal vascularity (*p* = 0.748), pseudo-capsule formation (*p* = 0.076), necrosis (*p* = 0.797), wash-out (*p* = 0.478), and tumor thrombus (*p* = 0.748), did not exhibit significant differences between RPHCC and control groups (Table 2). Similarly, MRI features also failed to show significant differences in internal vascularity (*p* = 0.582), pseudo-capsule formation (0.539), necrosis (*p* = 0.314), wash-out (0.722), and tumor thrombus (*p* = 1.000) (Table 2). CT irregular margins identified RPHCC with a sensitivity of 54.5% and a specificity of 100% (area under the curve, AUC = 0.773, *p* = 0.030).

### 3.2. Differences in Texture Features

There was a statistically significant difference across different CT texture features in spatial scale filters between the RPHCC and control groups, particularly for SSF-3 (medium-texture scale), SSF-4 (medium-texture scale), SSF-5 (medium texture scale), and SSF-6 (coarse-texture scale) (Table 3). Mean and mean of the positive pixels (MPP) were the most significant markers across the different filters (SSF-6—coarse-texture scale being “best”) and SD was close to significant at SSF-6 (coarse-texture scale) with a *p* = 0.056. We chose the coarse filter which was the best amongst other filters for significant metrics (mean and MPP) and close to the significant metric SD to develop a composite score that yielded clinically relevant diagnostic criteria.

The most significant differences were in SSF-6, coarse-texture scale, at mean *p* = 0.023, SD *p* = 0.056, MPP *p* = 0.023. The capacity of SSF-6 mean, SD, and MPP to identify RPHCC was evaluated, and mean SSF-6 < 16.57 identified RPHCC with a sensitivity of 90.1% and specificity of 63.6% (AUC = 0.785, *p* = 0.023), SD SSF-6 < 42.40 identified RPHCC with a sensitivity of 81.8% and specificity of 63.6% (AUC = 0.744, *p* = 0.053), and MPP SSF-6 < 33.73 identified the rapidly progressing HCC with a sensitivity of 81.8% and specificity of 63.6% (AUC = 0.785, *p* = 0.023). However, MRI texture feature analysis did not show significant differences across texture feature-extracted data in RPHCC and control groups (Table 4).

### 3.3. Diagnostic Criteria Based on Results of CT Texture Features

A composite diagnostic score was made by binarizing the four criteria and adding them together to RPHCC (RPHCC-1) from non-RPHCC (non-RPHCC-0). This criterion was used to develop a further composite score of mean SSF-6 binarized + SD SSF-6 binarized + MPP SSF-6 binarized + irregular margins. The Mann–Whitney test on this composite score was significantly different between rapidly progressing HCC and others (*p* = 0.001) with a higher score corresponding to RPHCC (Figure 5). A synergistic diagnostic score ≥ 3 identified the rapidly progressing HCC with a sensitivity of 81.8% and specificity of 81.8% (AUC = 0.884, *p* = 0.002) (Table 4).

## 4. Discussion

RPHCC is a subset of HCC that displays a heightened rate of tumor progression, which results in poor clinical outcomes and requires early clinical diagnosis for optimal management. Although liver transplantation can be curative in the early stages, there are strict tumor size restrictions that determine eligibility for transplant surgery [4]. Consequently, it is critical for RPHCC to be detected, differentiated from non-RPHCC tumors, and managed in its smaller stages at sizes when patients are still eligible for treatment. Since biopsies are no longer indicated prior to HCC treatment and since there are no known distinctive molecular features of RPHCC versus non-RPHCC, imaging serves as the primary method to identify tumors and determine the clinical course of treatment [15]. By delineating the visual features and the CT features that are able to diagnose RPHCC, we refined two independent, image-based, non-invasive methods of RPHCC identification. In doing so, we created a composite score of visual irregular margins and mean, SD, and MPP coarse-texture-scale CT features. With this score, we introduced the possibility of identifying RPHCC tumors at sizes when the patients are still eligible for transplantation if not even locoregional treatment.

Prior studies have correlated a higher histological grade and moderate or poor pathological differentiation with visually detected irregular margins on CT [16] and MRI [17]. Although the molecular and genomic profiles of RPHCC have not yet been characterized, lower tumor differentiation is generally associated with more aggressive phenotypes and progression [18]. In our results, visual radiologic features apart from irregular margins did not distinguish between RPHCC and non-RPHCC, suggesting that internal vascularity, pseudo-capsule formation, necrosis, wash-out, and tumor thrombus are not specific to the RPHCC subtype and do not uniquely contribute to a rapidly progressive HCC phenotype. With respect to these other visual radiologic features, a publication by Sugimoto et al. 2015 found that hypervascular HCC tumors can, in fact, be well-differentiated, and that there was no difference between observed tumor size and tumor markers between hypervascular and hypovascular HCC tumors [19]. Similarly, the presence of a pseudo-capsule in HCC tumors was not associated with a higher tumor grade or incidence of vascular invasion in a separate study published in 2009 [20].

CT texture analysis of HCC has been investigated in numerous contexts, including tumor grade and disease-free survival [14], survival after surgical resection [21], tumor heterogeneity [22], and transarterial chemoembolization [23]. As per our results, mean, SD, and MPP coarse CT texture features SSF-6 identified RPHCC with a sensitivity of 90.1%, 81.8%, and 81.8% respectively. Interestingly, MRI features did not distinguish RPHCC in our cohort, even though MRI provides a higher contrast resolution of soft tissues [24]. Our negative findings for MRI textures can be explained by the possibility that MRI features may need to be combined in order to be diagnostically meaningful, although individual features may not independently distinguish RPHCC tumors. To incorporate the same variables as CT, we looked at only contrast-enhanced features on MR and did not assess T2 weighted and diffusion imaging characteristics. For instance, in a study by Stocker et al., 2018, optimally derived combinations of MRI textures such as skewness, variance, kurtosis, and entropy were able to distinguish HCC from benign hepatocellular tumors in the non-cirrhotic liver [25]. Future studies should look into MRI texture combinations with an expanded cohort of RPHCC tumors and their controls.

A key limitation of our manuscript is in the size of the RPHCC cohort we analyzed. As a whole, RPHCC cases are limited in number for research studies, because the American Association for the Study of Liver Disease’s guidelines for the screening and removal of lesions often result in resection prior to tumor growth meeting RPHCC clinical diagnostic criteria. Validation of our features in an additional cohort should bolster the diagnostic capability of our binary diagnostic criterion. Further measures for validation include expanding inter-reader evaluations, which are limited in our current study. Indeed, multiple radiologists can corroborate the texture analysis and components that create the composite score for RPHCC. Histopathological correlations can further serve as markers of accuracy, although imaging features alone have clinical concordance independent of pathologic diagnosis [26]. Extensive research in RPHCC molecular and genomic characterization is necessary to develop biological associations with radiomic features and CT or MRI features.

## 5. Conclusions

Our current understanding of RPHCC is limited by cohort size and by an absence of comprehensive data regarding the specific imaging features that distinguish RPHCC from non-RPHCC tumors. We used visual radiologic features and texture analysis to create a diagnostic criterion to differentiate RPHCC from non-RPHCC tumors. Our results found that CT coarse-texture-scale SSF-6 features of mean SSF-6, SD SSF-6, and MPP SSF-6 in combination with visually detected irregular margins was able to statistically significantly differentiate between RPHCC and none-RPHCC. By developing an image-based, non-invasive diagnostic criterion, we created a composite score that can identify RPHCC patients at their early stages when they are still eligible for transplantation, improving the clinical course of patient care.

## Figures and Tables

**Figure 1 jpm-10-00136-f001:**
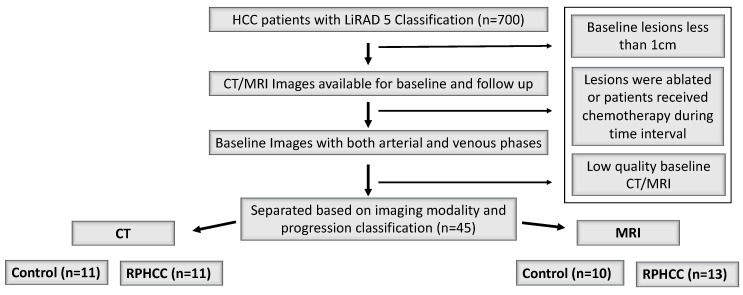
Inclusion/exclusion flowchart of patient selection for rapidly progressive hepatocellular carcinoma (RPHCC) and control tumors in the CT and MRI groups. CT cohort of n = 11 RPHCC and n = 11 non-RPHCC and an MRI cohort of n = 13 RPHCC and n = 10 non-RPHCLC.

**Figure 2 jpm-10-00136-f002:**
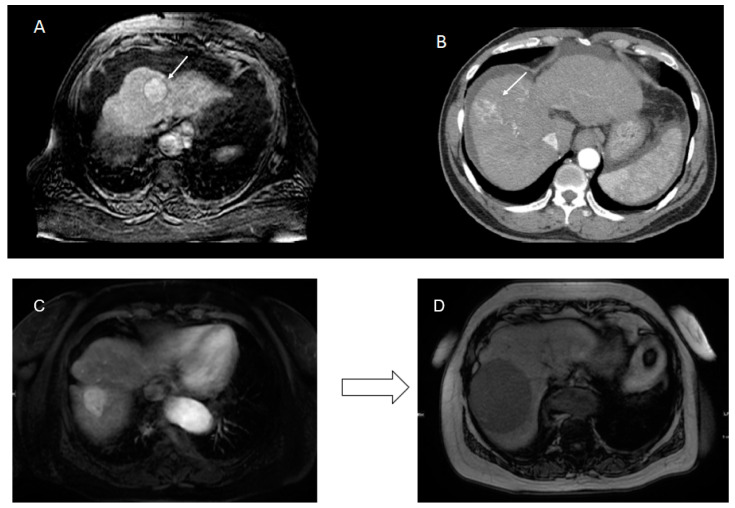
The included cases were separated based on imaging modality CT versus MRI and well-circumscribed (**A**) versus irregular margin (**B**); Example of rapidly progressive subtype of hepatocellular carcinoma progression in patient. Tumor increased in size from 3.3 (**C**) to 8.8 cm (**D**) in a period of 12 months, showing rapid growth in the same patient.

**Figure 3 jpm-10-00136-f003:**
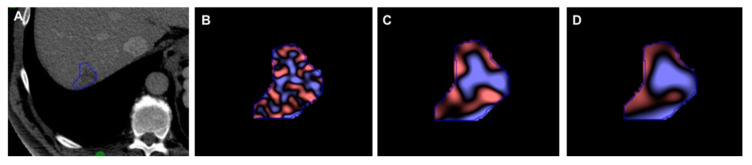
Representative single-slice images at the level of the largest diameter of the LIRADS 5 lesion in the arterial phase were identified and then sent to a commercially available texture analysis research software platform; then, texture features were extracted using commercially available research software. Using the software, a region of interest (ROI) was manually drawn on a single slice corresponding to the largest diameter of the lesion in the arterial phase for RPHCC (**A**) for (**B**) fine, (**C**) medium, and (**D**) coarse filters.

**Figure 4 jpm-10-00136-f004:**
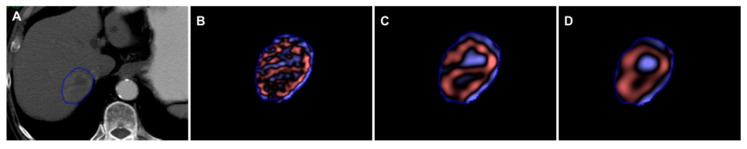
Representative single-slice images at the level of the largest diameter of the LIRADS 5 lesion in the arterial phase were identified and then sent to a commercially available texture analysis research software platform; then, texture features were extracted using commercially available research software. Using the software, a region of interest (ROI) was manually drawn on a single slice corresponding to the largest diameter of the lesion in the arterial phase for non-RPHCC (**A**) for (**B**) fine, (**C**) medium, and (**D**) coarse filters.

**Figure 5 jpm-10-00136-f005:**
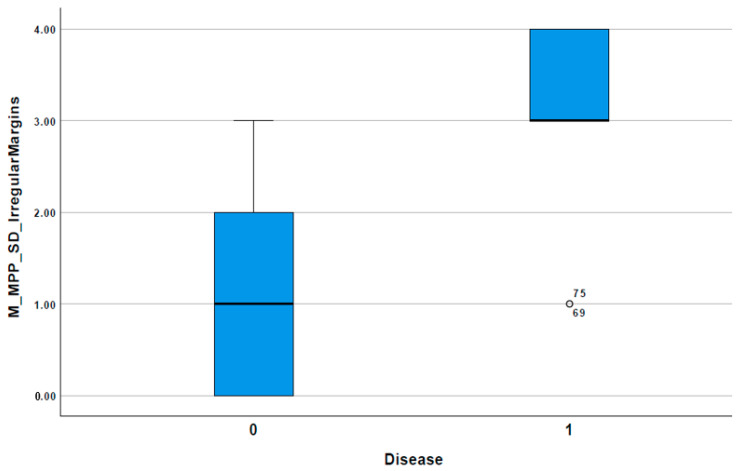
Composite score ≥3 identified the rapidly progressing HCC with a sensitivity of 81.8% and specificity of 81.8% (AUC = 0.884, *p* = 0.002).

**Table 1 jpm-10-00136-t001:** Demographic data, comorbidities, Child-Pugh Class, and mean MELD score. None of the factors reached statistical significance between RPHCC and control tumors at *p* < 0.05.

	RPHCC (n = 24)	Control (n = 21)	All (n = 45)
Demographics			
Age (range)	64	69.48	66.56
%Male	75.00%	76.19%	75.56%
Caucasian	37.50%	47.62%	42.22%
African American	25.00%	14.29%	20.00%
Other	37.50%	38.10%	37.78%
Comorbidities			
Hep B	4.17%	9.52%	6.67%
Hep C	50.00%	52.38%	51.11%
Cirrhosis	66.67%	76.19%	71.11%
NASH	16.67%	9.52%	13.33%
Diabetes	54.17%	42.86%	48.89%
Chronic Kidney Disease	8.33%	9.52%	8.89%
Child-Pugh Class			
Class A	54.17%	47.62%	51.11%
Class B	41.67%	52.38%	46.67%
Class C	4.17%	0.00%	2.22%
Mean MELD Score	10.53	10.57	10.55

**Table 2 jpm-10-00136-t002:** Mean analysis of clinical and imaging variables for CT and MR RPHCC and control groups. *p*-values obtained by the Mann–Whitney U-test, indicating that CT irregular margins are visually different between the RPHCC and non-RPHCC patients at *p* = 0.028. * indicates a statistically significant difference in texture analysis feature extraction value between groups using the Mann–Whitney test.

Variable	CT-RPHCC (n = 11)	CT-Control (n = 11)	CT *p*-Value	MR-RPHCC (n = 13)	MR-Control (n = 10)	MR *p*-Value
Irregular Margins	54.55%	0%	0.028 *	23.07%	0%	0.539
Necrotic Component	54.55%	5.45%	0.797	22.69%	9.0%	0.314
Internal Vascularity	27.27%	36.36%	0.748	30.7%	10.0%	0.582
Wash-Out	81.81%	100%	0.478	100%	90.0%	0.722
Pseudo-Capsule	18.18%	63.63%	0.076	61.53%	50.0%	0.539
Tumor Thrombus	9.01%	0%	0.748	0%	0%	1.000
Baseline Lesion Size	3.37 cm	3.58 cm	0.193	2.45 cm	2.66 cm	0.722
AFP > 10	42.85%	60.0%	0.755	62.5%	57.14%	0.867

**Table 3 jpm-10-00136-t003:** Analysis of p-values for CT and MR texture features across two groups. * indicates a statistically significant difference in texture analysis feature extraction value between groups using the Mann–Whitney test.

Variable	CT	CT	CT	CT	CT	CT	MR	MR	MR	MR	MR	MR
	SSF-0	SSF-2	SSF-3	SSF-4	SSF-5	SSF-6	SSF-0	SSF-2	SSF-3	SSF-4	SSF-5	SSF-6
Mean	0.171	0.056	0.028 *	0.040 *	0.023 *	0.023 *	0.346	0.821	0.923	0.974	0.974	0.974
SD	0.606	0.478	0.217	0.151	0.133	0.056	0.628	0.722	0.821	0.771	0.771	0.974
Entropy	0.606	0.562	0.27	0.193	0.116	0.076	0.974	1	0.872	0.772	0.674	0.582
MPP	0.133	0.562	0.133	0.076	0.040 *	0.023 *	0.346	0.821	0.771	0.923	1	0.974
Skewness	0.193	0.652	0.562	0.898	0.898	0.847	0.08	0.456	0.497	0.582	0.346	0.346
Kurtosis	0.652	0.606	0.3	0.401	0.332	0.478	0.159	0.418	0.203	0.228	0.283	0.381
Total	0.748	0.748	0.748	0.748	0.748	0.748	0.381	0.381	0.381	0.381	0.381	0.381

**Table 4 jpm-10-00136-t004:** Identification of RPHCC. Diagnostic criteria based on results from CT texture features. Mann–Whitney analysis of sensitivity, specificity, area under the curve (AUC) and *p*-value.

Criteria	Sensitivity	Specificity	AUC	*p*-Value
Mean_SSF6 < 16.57	90.1%	63.6%	0.785	0.023
MPP_SSF6 < 33.73	81.8%	63.6%	0.785	0.023
SD_SSF6 < 42.40	81.8%	63.6%	0.744	0.053
Irregular Margins	54.5%	100%	0.773	0.030
Binary Diagnostic Composite Score ≥ 3	81.8%	81.8%	0.884	0.002

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
