# Peer review of "Retrospective CT/MRI Texture Analysis of Rapidly Progressive Hepatocellular Carcinoma"

_jpm, 2020, doi:10.3390/jpm10030136_

Round 1

Reviewer 1 Report

#This article entitled as “ Retrospective CT/MRI Texture Analysis of Rapidly Progressive Hepatocellular Carcinoma (HCC)”, was to evaluate the effectiveness of texture analysis using CT and MRI images to discriminate the rapidly progressive HCC (RPHCC) from non-rapidly progressive HCC(NRPHCC). This study has significant limitation as follow: small number of study population (RPHCC: NRPHCC= 24:21) with CT (11:11) and MRI (14:11) images, no validation data for results from texture analysis.

#1 Tumor characteristics, such as tumor size, number of tumor(s), vascular invasion, etc., could influence the progression of tumor(s). Thus, these tumor characteristics should be compared between RPHCC and NRPHCC.

#2 In this study, rapid progression was defined as 50% growth in <3 months, 100% growth in <6 months, 150% growth in <12 months, or 200% growth in <15 months. It would be better to describe the measurement of tumor measurement for rapid progression, i.e., sum of largest diameters of tumor(s) or sum of each volume(s) tumor(s).

#3. When use rapid progression criteria, did patients of RPHCC and NRPHCC have different prognosis? The purpose of this study was accurate detection of RPHCC and subsequent improvement of clinical outcomes. Thus, clinical outcomes, such as overall survival, according to rapid progression criteria should be needed.

#4 Combination of traditional image findings , such as irregular margin, necrotic components, internal vascularity, washout, etc., showed in Table 2, are also likely to influence the rapidly progression of HCC. Thus, it would be better to compare the texture analysis with combination of traditional image findings.

Reviewer 2 Report

To the authors:

“Retrospective CT/MRI Texture Analysis of Rapidly Progressive Hepatocellular Carcinoma”

My comments are as follows.

  1. Please correct “Child-Pugh grade” to “Child-Pugh class”.
  2. In Abstract, you described “a CT cohort of n=11 RPHCC and n=11 non-RPHCC and an MRI cohort of n=13 RPHCC and n=10 non-RHPCC”. Are the numbers correct in Figure 1?
  3. You mentioned “CT coarse-texture-scale features in combination with visually detected irregular margins was able to statistically differentiate between RPHCC and non-RPHCC.” How did you define “irregular margins”?
  4. Please add HCC tumor marker, serum des-gamma-carboxy prothrombin (DCP), to the variable (Table.2).

I hope that my comments will be useful in improving the article.

Round 2

Reviewer 1 Report

None.

Reviewer 2 Report

The manuscript has been revised well.